# Intestinal Transcytosis of a Protein Cargo and Nanoparticles Mediated by a Non-Toxic Form of *Pseudomonas aeruginosa* Exotoxin A

**DOI:** 10.3390/pharmaceutics13081171

**Published:** 2021-07-29

**Authors:** Ruiying Li, Floriane Laurent, Alistair Taverner, Julia Mackay, Paul A. De Bank, Randall J. Mrsny

**Affiliations:** Department of Pharmacy and Pharmacology, University of Bath, Bath BA2 7AY, UK; ruiyingli1991@hotmail.com (R.L.); florine_laurent@hotmail.com (F.L.); at420@bath.ac.uk (A.T.); jb281@bath.ac.uk (J.M.); pd277@bath.ac.uk (P.A.D.B.)

**Keywords:** oral protein delivery, in vivo model, transcytosis, nanoparticle

## Abstract

The low permeability of nanoparticles (NPs) across the intestinal epithelium remains a major challenge for their application of delivering macromolecular therapeutic agents via the oral route. Previous studies have demonstrated the epithelial transcytosis capacity of a non-toxic version of *Pseudomonas aeruginosa* exotoxin A (ntPE). Here, we show that ntPE can be used to deliver the protein cargo green fluorescent protein (GFP) or human growth hormone (hGH), as genetic fusions, across intact rat jejunum in a model where the material is administered by direct intra-luminal injection (ILI) in vivo in a transcytosis process that required less than 15 min. Next, ntPE chemically coupled onto biodegradable alginate/chitosan condensate nanoparticles (AC NPs-ntPE) were shown to transport similarly to ntPE-GFP and ntPE-hGH across rat jejunum. Finally, AC NPs-ntPE loaded with GFP as a model cargo were demonstrated to undergo a similar transcytosis process that resulted in GFP being colocalized with CD11c^+^ cells in the lamina propria after 30 min. Control NP preparations, not decorated with ntPE, were not observed within polarized epithelial cells or within the cells of the lamina propria. These studies demonstrate the capacity of ntPE to facilitate the transcytosis of a covalently associated protein cargo as well as a biodegradable NP that can undergo transcytosis across the intestinal epithelium to deliver a noncovalently associated protein cargo. In sum, these studies support the potential applications of ntPE to facilitate the oral delivery of macromolecular therapeutics under conditions of covalent or non-covalent association.

## 1. Introduction

The advantages of delivering macromolecular therapeutic agents (i.e., biopharmaceuticals such as proteins and polynucleic acids) via the oral route over the currently used injection routes are well accepted [1]. Despite this preference and nearly a century of research, no methodology has been identified for the efficient, safe, and consistent delivery of biopharmaceuticals via the oral route. The recent approval of an oral semaglutide tablet by the FDA, however, has provided optimism for at least the oral delivery of therapeutic peptides, although the bioavailability appears to be less than 1% and quite variable [2,3]. Unfortunately, methods to enhance the uptake of peptide therapeutics across intestinal epithelia do not appear viable for large biopharmaceuticals due to differences in physicochemical properties and/or the requirement of retaining a more complex native 3D structure. Although the hostile environs of the stomach can be avoided using enteric coated dosage forms, the low permeability of biopharmaceuticals across the intestinal epithelium remains as a limiting barrier [4,5]. In the case of peptides, two routes across the epithelium can potentially be used: the paracellular route between adjacent epithelial cells such as the mechanism stimulated by PIP peptides [6] and transcellular transport in a manner similar to small molecule uptake, which is facilitated by the permeation enhancer, salcaprozate sodium, present in the semaglutide tablet [7].

A sufficient opening of the paracellular route to support the passage of large biopharmaceuticals, however, carries the risk of introducing unintended substances into the body, potentially causing untoward inflammatory outcomes [8] and the large molecular size, hydrophilicity and requirement for retention of a native 3D structure of proteins limit the potential for strategies to enhance transcellular uptake. Thus, transcytosis following apical endocytosis and subsequent vesicular trafficking provides the most feasible route across the epithelium for therapeutic proteins [9]. This transcytosis approach has been explored experimentally for non-selective vesicular uptake, which predominantly occurs at the luminal surface of microfold (M) cells [10]. As the number of M cells in the human intestine is extremely limited and restricted to selected sites, this approach has not been clinically advanced for oral delivery of biopharmaceuticals. 

While there are a number of cell types that make up the intestinal epithelium, enterocytes far outnumber all the others [11]. A transcytosis pathway across enterocytes has previously been demonstrated for the exotoxin A (PE) virulence factor secreted by *Pseudomonas aeruginosa*; a single amino acid deletion in the catalytic domain of PE (deletion of aspartic acid at position 553; ΔE553) renders the protein non-toxic (ntPE) but allows it to retain its transcytosis capacity [12]. Since incorporating biopharmaceuticals into nanoparticles (NPs) has been shown to prevent their destruction in the intestinal lumen [13], we postulated that NP transport across the intestinal epithelium could be enhanced by their decoration with ntPE. This possibility was examined in rat jejunum in vivo using biodegradable NPs prepared as an alginate/chitosan condensate (AC NPs-ntPE) with green fluorescent protein (GFP) as a model cargo. Our results provide support for ntPE-directed apical endocytosis and intracellular trafficking to enhance the transcytosis of NPs that could carry a biopharmaceutical payload.

## 2. Materials and Methods

### 2.1. Preparation of ntPE-TEV-H_6_

A non-toxic ΔE553 mutant of *Pseudomonas aeruginosa* exotoxin A (ntPE) was modified at the C-terminus of the open reading frame to include a consensus sequence for the tobacco etch virus (TEV) protease placed between two cysteine residues and followed by a hexa-histidine sequence (CENLYFQSGTCHHHHHH): ntPE-TEV-H_6_, was kindly provided by Dr. David FitzGerald (NCI/NIH, Bethesda, MD, USA). SHuffle^®^ T7 *E. coli* (New England BioLabs, Ipswich, MA, USA) transformed with were induced using 0.5 mM isopropyl β-D-1-thiogalactopyranoside (Fisher Scientific, Loughborough, UK). Cells were harvested by centrifugation at 1530× *g* at 4 °C for 20 min and suspended in 30 mM Tris-HCl (Sigma-Aldrich, Gillinghamm, UK), 20% sucrose (Sigma-Aldrich, Gillinghamm, UK) at pH 8.0, with 200 mM EDTA (Sigma-Aldrich, Gillinghamm, UK) added dropwise on ice until a final concentration of 1 mM was reached. Following 10 min of gentle shaking, cell suspensions were centrifuged at 3225× *g* for 20 min at 4 °C, with pellets suspended in ice-cold water and incubated on ice with gentle shaking for 10 min. Following this osmotic shock step, cell debris was pelleted by centrifugation at 3225× *g* for 20 min at 4 °C and the supernatant, containing proteins from the periplasm, was collected. The ntPE-TEV-H6 protein was captured using a 5 mL prepacked HisTrap HP column (GE Healthcare, Abingdon, UK) and eluted using a linear 20–500 mM gradient of imidazole (Acros Organics, NJ, USA) in a 50 mM Tris pH 8.0 buffer containing 300 mM sodium chloride. Fractions containing the ntPE-TEV-H6 protein were pooled and concentrated using a 10 kDa MWCO Amicon^®^ Ultra-4 Centrifuge Filter (Merck Millipore Ltd., Darmstadt, Germany) prior to size-exclusion chromatography using a Superdex^®^ 200 HR10/30 column (GE Healthcare, Chicago, IL, USA) using PBS at a flow rate of 0.5 mL/min. SDS-PAGE was used to identify column fractions containing ntPE-TEV-H6. Protein concentrations were determined using a NanoDrop™ 2000c spectrophotometer (Thermo Scientific, Wilmington, DE, USA) at 280 nm.

### 2.2. Preparation of ntPE-GFP and ntPE-hGH

Chimeras of ntPE linked at its C-terminus to the N-terminus of either green fluorescent protein (GFP) or human growth hormone were prepared by the genetic fusion of the non-toxic ΔE553 mutant of *Pseudomonas aeruginosa* exotoxin A using expression and purification previously described for this family of toxins [14]. 

### 2.3. Preparation of ntPE for Decorating Nanoparticles (NPs)

First, 2 mg of ntPE-TEV-H6 protein was reacted with TEV enzyme (Life Technologies, Carlsbad, CA, USA) and a 1 mL HisTrap HP column was used to elute the cleaved protein (ntPE-CENLYFQ). A 5-fold molar excess of *N*-(ε-maleimidocaproic acid) hydrazide (EMCH) was reacted with ntPE at RT for 2 h prior to the removal of unreacted EMCH using dialysis (10 kDa MWCO) against PBS overnight at 4 °C. To prepare a control protein for coupling to NPs, a 3-fold molar excess of Traut’s reagent (5,5′-dithiobis(2-nitrobenzoic acid); Sigma-Aldrich, Gillinghamm, UK) was added to BSA in 12 mM PBS (pH 7.4) containing 3 mM EDTA and incubated at RT for 1 h to produce ~1 thiol group per BSA molecule prior to reaction with EMCH in a manner identical to that used for ntPE-CENLYFQ.

### 2.4. Preparation of Alginate/Chitosan NPs

First, 2 mL of 0.51 mg/mL calcium chloride (Sigma-Aldrich, Gillinghamm, UK) solution was added to 10 mL of 0.06 mg/mL oxidized alginate solution dropwise under micro-tip probe ultra-sonication (Branson Sonifier Model 2501450, Branson, CT, USA). The resultant material was stirred for another 30 min before the addition of 2 mL of a 0.3 mg/mL aqueous chitosan solution (Sigma-Aldrich, UK) and the suspension equilibrated overnight at RT to allow the formation of NPs. The material was reacted with BSA-EMCH or ntPE-EMCH: 2 mL of these NPs were reacted with 0.5 mg of BSA-EMCH or ntPE-EMCH for 15 min at RT prior to the addition of 4 mg of EDC (1-ethyl-3-(3-dimethylaminopropyl) carbodiimide, Sigma-Aldrich, Gillinghamm, UK) and 5 mg of NHS (*N*-hydroxysuccinimide, Thermo Fisher Scientific, Runcorn, UK) to the reaction solution (Figure 1). After the addition of EDC and NHS, the pH was adjusted to 6.5 and the reaction was allowed to proceed at RT. After 2 h, unreacted protein and reagents were removed by overnight dialysis against water at RT (100 kDa, Spectra/Por^®^ Biotech membranes, SpectrumLabs, New Brunswick, NJ, USA). The extent of protein coupling onto AC NPs was quantified using the Bradford assay ((Bio-Rad Laboratories, Hercules, CA, USA). After measuring the extent of labelling with ntPE or BSA, remaining aldehyde groups were reacted with an excess of the fluorescent dye Alexa Fluor^®^ 546 (Ex/Em 490 nm/525 nm) to provide a label for visualization by fluorescence microscopy. GFP-loaded AC NPs were prepared by mixing 0.3 mg of GFP (prepared in lab following expression in *E.coli* [15]) with 2 mL of 0.3 mg/mL chitosan solution, and this mixture was added dropwise into the alginate–calcium complex.

### 2.5. Measurement of Nanoparticle Size and Zeta Potential

Zeta potential measurements were performed in triplicate using a Malvern ZetaSizer Nano, (Malvern Instruments, Worcestershire, UK) at 25 °C. The hydrodynamic diameter of NPs and particle concentration was measured using a Malvern NanoSight NS 500 instrument, (with all samples diluted 10-fold in PBS prior to measurement. Instrument settings were standardized for measuring each type of NP, with three recordings obtained for each sample. Protein density on each NP preparation was calculated as follows:(1)$$protein\;density\;\left(molecules\;per\;particle\right)=\frac{c\times 10^{6}\times N}{M\times cp}$$
with *c* being the concentration of protein on NPs in 1 mL, *M* being protein molecular weight, *cp* being the number of nanoparticles in 1 mL, and *N* being Avogadro′s number.

### 2.6. Transmission Electron Microscopy (TEM)

A morphological analysis of NPs was performed using a transmission electron microscope (TEM; JEOL JEM1200EXII, Jeol, Tokyo, Japan). NP suspensions were diluted 10-fold in PBS (pH 7.4) and dropped onto carbon-coated copper grids (FC300Cu, EM Resolutions, Saffron Walden, UK) with the liquid being quickly removed by touching the edge of the grid with filter paper. Negative staining was performed by exposing samples to 2% uranyl acetate for 30 s, which was also removed by filter paper absorption. Samples were stored in a desiccator after air drying until being viewed in the TEM. At least three batches of each NP were prepared. Representative TEM photos presented in the manuscript were taken from batches prepared on the same date to maximize comparing/characterize NPs. Average NP size was determined from the measurement of 100 randomly selected NPs from different locations on the samples.

### 2.7. Fourier Transform Infrared Spectroscopy of Alginate/Chitosan NPs

FTIR spectra were obtained by using a Perkin Elmer Frontier Optica FTIR spectrometer (Frontier, Perkin Elmer Ltd, Cambridge, UK) equipped with a mercury cadmium telluride detector. Buffer correction was carried out before each measurement. A drop of AC NPs, AC NPs-BSA and AC NPs-ntPE suspension was placed on the crystal and signals were obtained from 4000 to 600 cm^−1^ wavenumbers at a resolution of 4 cm^−1^.

### 2.8. Oxidation of Alginate

Oxidized alginate was prepared by periodate oxidation [16]. Briefly, a 1% (*w*/*v*) sodium alginate (W201502, Sigma-Aldrich, viscosity = 5.0–40.0 cps at 1% in water at RT) solution was mixed with 50 mM sodium periodate (Sigma-Aldrich, Gillinghamm, UK) and incubated at RT for 24 h before quenching with an equimolar amount of ethylene glycol (Sigma-Aldrich, Gillinghamm, UK). Sodium chloride (2.5 g) was added, followed by precipitation with 2 volumes of ethyl alcohol (200 mL). The precipitate was collected by centrifugation and dissolved in 100 mL distilled water prior to a second precipitation using 200 mL ethanol. The resulting product was freeze-dried to yield a white solid (0.8 g, 80% yield).

### 2.9. In Vivo Transcytosis Assay Protocol

An in vivo protocol, referred as the intraluminal (ILI) injection model, to assess the capacity of test articles to transport across intact rat jejunal epithelium was performed as previously published [14]. Briefly, male Wistar rats (250–300 g, bred in-house) were anesthetized using isoflurane and a 4–5 cm midline abdominal incision was used to expose the jejunal region of the small intestine. Test articles of ntPE-GFP, AC NPs (AC NPs-ntPE, AC NPs-BSA) or GFP-loaded AC NPs (GFP-AC NPs-ntPE, GFP-AC NPs-BSA), were all administered at 86 µg/mL, were suspended in 250 μL PBS and slowly (~20–30 s) injected into the lumen. Mesentery adjacent to the injection site was denoted with using a permanent marker. At study termination, a 3–5 mm region that captured the marked intestine segment was isolated and processed for microscopic assessment [14].

### 2.10. Immunofluorescent Microscopy

Isolated intestinal tissues were fixed in 4% paraformaldehyde at 4 °C for 18–24 h, processed using a Leica TP1020 tissue processor, dehydrated in increasing concentrations of ethanol, cleared with HistoClear (National Diagnostics, Ason Clinton, UK) and infused with molten paraffin wax. Sections cut from tissue-embedded paraffin wax blocks (5 mm thickness; Jung Biocut2035 microtome) were mounted on glass microscope slides, rehydrated, and processed for antigen retrieval by boiling slides in 10 mM sodium citrate for 10 min followed by washing with PBS [14]. Tissue sections were then permeabilized for 30 min at RT using 0.2% Triton X-100 in PBS and incubated at RT for 2 h in blocking buffer (2% BSA; 2% donkey serum, Sigma-Aldrich; and 0.1% Triton-X 100 in PBS) prior to incubation with primary antibody (rabbit polyclonal anti-*Pseudomonas aeruginosa* exotoxin A prepared by our lab, diluted 1:1000) in PBS containing 0.1% Triton X-100 and 1% BSA overnight at 4 °C. GFP was detected with a rabbit polyclonal antibody (Abcam; Ab290) and human growth hormone was detected with a goat polyclonal antibody (R&D systems; AF1067). Excess primary antibody was removed by rinsing in PBS prior to incubation at RT for 2 h in a secondary antibody solution (Alexa Fluor^®®^ 546-conjugated donkey anti-rabbit polyclonal IgG, diluted 1:100, A10040, Life Technologies, Paisley, UK) containing 0.1% Triton X-100 in PBS. Tissue sections were incubated for 1 h with 200 nM DAPI (4′, 6-diamino-2-phenylindole, Dihydrochloride, D1306, Thermo Fisher Scientific, Runcorn, UK) prior to analysis using a Zeiss LSM 510 microscope. DAPI was detected with an excitation wavelength was 405 nm and the emission wavelength was 462 nm. CD11c^+^ cells were detected using the same immunohistochemistry staining as AC NPs using a mouse monoclonal anti-CD11c antibody in the primary antibody solution and Alexa Fluor^®®^ 488 (Ex/Em 490 nm/525 nm, ab150109, Abcam, diluted 1:100 containing 0.1% Triton X-100 in PBS) conjugated donkey anti-mouse polyclonal IgG in the secondary antibody solution.

## 3. Results

### 3.1. Conjugation of ntPE or BSA onto NPs

At present, it is unclear what components of ntPE are required for its transport across enterocytes. In order to optimize the retention of this function after coupling to NPs, a construct was designed that could be used for selective chemical conjugation through the C-terminal segment of domain III, whose catalytic function is not required for transcytosis [12]. A tobacco etch virus (TEV) protease-specific sequence (CENLYFQ√SGTC, where √designates the amide bond cleaved by the TEV protease) followed by a hexa-histidine sequence (H_6_) were introduced into the pVC 45D plasmid at the C-terminus of the ntPE open reading frame to produce ntPE-CENLYFQSGTCHHHHHH (ntPE-TEV-H_6_). The ntPE-TEV-H_6_ construct, expressed in *E. coli* and purified using a two-step process, provided a protein of the anticipated molecular weight (theoretical = 71.2 kDa) with a purity of >90% (Figure 2). Using a heterobifunctional coupling reagent, EMCH, the free sulfhydryl residue of ntPE-CENLYFQ was conjugated to carboxylate groups on the surface of the AC NPs (Figure 3). After introducing a free sulfhydryl on bovine serum albumin (BSA-SH) using Traut’s reagent, a similar EMCH-mediated coupling to AC NPs was performed (Figure 3).

### 3.2. Characterization of GFP-Containing AC NPs Coupled to ntPE or BSA

The biodegradable AC NPs were characterized by TEM and determined to have an average diameter of 48 ± 14 nm in the dehydrated state (Table 1, Figure 4). After conjugation with BSA-SH or ntPE-CENLYFQ, TEM analysis demonstrated that the particle diameters in the dehydrated state were similar to the unmodified particles: AC NPs-BSA had an average diameter of 48 ± 15 nm and AC NPs-ntPE had an average diameter of 46 ± 24 nm (Table 1; Figure 4). There was no suggestion of extensive aggregation for AC NPs-BSA or AC NPs-ntPE preparations examined under these conditions. While previous studies have suggested that the presence of proteins on the surface AC NPs might shield them from self-association by increasing their colloidal stability [17], this possibility was not experimentally tested in our studies. Particle size for AC NPs remained unchanged for 7 days in the simulated intestinal fluid (SIF, pH 6.8), the TEM images for AC NPs and NPs-PE was captured after dispersing them in SIF (pH 5.0, 6.8, or 7.0) for 15, 30, or 60 min. Under all these conditions, NPs were observed by TEM.

Hydrated AC NPs had an average diameter of 215 ± 8 nm when measured using DLS (Table 1). This diameter increased to 280 ± 9 nm after chemical coupling of BSA-SH but appeared to decrease slightly to 202 ± 20 nm after conjugation with ntPE-CENLYFQ (Table 1). As assessed by Student’s T test, hydrated blank AC NPs diameters were not statistically different from AC NPs-ntPE, however, AC NPs BSA diameters were different from both blank AC NPs and AC NPs-ntPE (0.003; *p* < 0.1). The zeta potential for hydrated AC NPs was +20 ± 8 mV, with the overall positive charge being attributed to the presence of chitosan. After chemical coupling with proteins, the zeta potential dropped to −19 ± 11 and −6 ± 5 mV for AC NPs-BSA and AC NPs-ntPE preparations, respectively (Table 1). All three of the NPs described in Table 1 were fully dispersed and remained in suspension for the duration of the characterization and administration studies performed subsequently. Based upon the calculated isoelectric point (pI) values of 5.82 for BSA and 5.31 for ntPE, these shifts toward a more negatively charged surface was expected. Using the number of particles/mL in these AC NPs, the number of BSA or ntPE molecules per nanoparticle was calculated. An average number of 3.30 ± 0.90 × 10^5^ protein molecules on each AC NP-BSA and 3.41 ± 0.82 × 10^5^ protein molecules on each AC NP-ntPE was estimated (Table 1).

The dehydration of AC NPs as a processing step for TEM studies emulates the potential for these materials to be prepared in the dry state that would be required for their ultimate application in a solid oral dosage form. While there was relatively large variation in these negative-stained “primary particles”, there was sufficient uniformity of the preparations that had a protein coating (AC NPs-BSA and AC NPs-ntPE) as their dehydration allowed them to remain as individual particles. This property bodes well for a potential solid oral dosage formulation of these materials. The measured hydrodynamic size of these AC NPs in solution places them in the desired range to fit inside intracellular vesicles that should participate in the receptor-mediated transcytosis events mediated by ntPE [12].

FTIR (Fourier-transform infrared spectroscopy) was next used to characterize the secondary structure of the BSA or ntPE associated with AC NPs prepared for these studies [18]. AC NPs before covalent coupling with BSA or ntPE showed strong intermolecular hydrogen bonds between alginate and chitosan at the broad peak of 3700–3000 cm^−1^ (Figure 5), consistent with a previous study [19]. FTIR spectra obtained for AC NPs-BSA and AC NPs-ntPE preparations demonstrated multiple bands typical of proteins, in particular stretches characteristic of amide bonds (1650 cm^−1^), indicating the presence of these proteins that would be consistent with their covalent attachment to AC NPs. Further, peaks observed at 1020 and 1030 were consistent with the presence of alginate and chitosan [20]. Together, these FTIR spectra support the predicted composition of AC NPs before covalent coupling with BSA or ntPE.

### 3.3. In Vivo Transcytosis of AC NPs Coupled to ntPE or BSA

Previous studies have demonstrated the potential for a biotinylated form of ntPE to transport across intact mouse nasal epithelium in vivo [21]. In order to test the hypothesis that ntPE could facilitate the transcytosis of a nanoparticle across intestinal mucosa, we first examined the potential of ntPE to ferry a macromolecule across intact intestinal epithelium in vivo using the intraluminal intestinal (ILI) model. A genetic fusion of ntPE with green fluorescent protein (ntPE-GFP) was observed to transport across intact intestinal epithelium and begin to accumulate in cells within the intestinal lamina propria by ~15 min after ILI into rat jejunum (Figure 6a). Following administration, ntPE-GFP was observed in vesicles present in the apical and basal regions of enterocytes, consistent with a transcytosis process of transport across the epithelium. Additionally, ntPE-GFP was detected in some of the non-polarized cells within the lamina propria, consistent with directed targeting to selected cells within this tissue compartment. We repeated these studies with another chimera where ntPE was genetically conjoined to human growth hormone (ntPE-hGH). ILI of ntPE-hGH at ~15 min showed a pattern of transport across the epithelium and accumulation within the lamina propria similar to that observed for ntPE-GFP (Figure 6b).

AC NPs prepared for these studies showed sufficient in vitro stability to allow for size determination, zeta potential measurement, and FTIR analysis. Thus, we believed that they would be sufficiently stable for in vivo testing and tested their uptake across rat intestinal epithelia using an intraluminal injection (ILI) method. Fluorescence microscopy was used to follow Alexa Fluor^®®^ 546 that had been chemical coupled to ntPE prior to its conjugation onto the surface of AC NPs (AC NPs-ntPE*). At 15 min after intraluminal injection into rat jejunum, intestinal tissues were isolated and processed for microscopy following PBS washing to remove adherent luminal surface materials. At this time point, an extensive uptake of AC NPs-ntPE* into enterocytes and goblet cells was observed (Figure 7a,b). The cellular distribution of AC NPs-ntPE*) was similar to the previous observations that examined a biotinylated form of ntPE to transport across intact mouse nasal epithelium in vivo [21] and ntPE genetically linked to hGH (Figure 6a).

By 30 min, the presence of AC NPs-ntPE* in enterocytes and goblet cells had diminished greatly, with residual Alexa Fluor^®®^ 546 label being primarily detected in non-polarized cells within the *lamina propria* (Figure 7c,d). These results are consistent with the possibility that these biodegradable particles are broken down soon after the completion of transcytosis following their transcytosis and entry into non-polarized cells within the lamina propria as previously described for this class of exotoxins [14]. The loss of structures containing ntPE* in a focused format (associated with AC NPs) would result a dramatic reduction in detectable fluorescence, with only sites where the fluorescent label is collected, possibly in lysosomes remaining detectable using this microscopic technique. 

Studies in this in vivo model suggested a rapid transcytosis capacity for AC NPs-ntPE* across the rat jejunal intestinal epithelium that resulted in uptake by selected non-polarized cells within the underlying lamina propria. Notably, AC NPs-ntPE* fate in the lamina propria appeared comparable to that previously observed for native, toxic PE uptake into macrophages and/or dendritic cells [22]. An ILI study into rat jejunum was also performed where BSA was labelled with Alexa Fluor^®®^ 546 prior to its covalent coupling to AC NPs (AC NPs-BSA*) and administered using the ILI procedure. In this case, AC NPs-BSA* were not observed in polarized cells of the epithelium or within the non-polarized cells of the lamina propria (data not shown). Together, these data suggest that ntPE-directed transcytosis of AC NPs did not coincide with non-selective uptake mechanisms that has been observed for the non-specific uptake of particulate and soluble antigens [23], but was consistent with a rapid and efficient transcytosis that would be expected with targeted and selective receptor-mediated vesicular trafficking.

We next examined the feasibility of using AC NPs-ntPE, lacking a fluorescent label on the ntPE, to deliver a protein cargo across the intestinal epithelium and into the lamina propria. GFP was selected as a model protein to allow for direct comparison with our previous in vivo findings and to determine if the pathway used by AC NPs-ntPE intersected with proteolytic compartments; extensive proteolysis would result in a loss of GFP’s fluorescent signature. GFP was incorporated into AC NPs at the time of coacervation with an encapsulation efficiency of ~92%. In vivo trans-epithelial transport studies of GFP-loaded AC NPs-ntPE or GFP-loaded AC NPs-BSA were conducted as before but now focused on a time point where these particles would have completed transcytosis and been sequestered into cells in the lamina propria. Rat jejunal intestinal tissue isolated 30 min following ILI of AC NPs-ntPE-GFP demonstrated the presence of GFP^+^ particles transporting across the epithelium and in selected cells within the lamina propria (Figure 8a). Animals dosed similarly for 30 min with AC NPs-BSA-GFP loaded with GFP demonstrated the sporadic presence of GFP at the luminal surface of the epithelium or within apical regions of epithelial cells but not in a manner that suggested epithelial transcytosis or delivery to cells within the lamina propria (Figure 8b).

Considering the known cell targeting specificity of native, toxic PE [22], AC NPs-ntPE-based delivery would be anticipated to target to macrophages and/or dendritic cells within the lamina propria that would express CD11c. At 30 min post-ILI, AC NPs-ntPE-GFP was observed in cells within the epithelium and in cells within the lamina propria that were CD11c+. The distribution of GFP^+^/CD11c^+^ cells within the lamina propria were consistent with dendritic cells and/or cells of myeloid lineage [24]. Hence, it appears that the fate of a protein cargo delivered by AC NPs-ntPE to the lamina propria would, at least partially, involve targeted uptake by cells that could be involved in antigen presentation.

## 4. Discussion

One of the greatest challenges to overcoming the barriers of efficient oral protein delivery is improving the vanishingly small transport capacity of biopharmaceuticals across the intestinal epithelium. We have addressed this challenge by using a bacterial toxin known for its transcytosis properties with the goal of increasing the transcytosis capacity of nanoparticles (NPs). In the studies presented, we show that a non-toxic version of exotoxin A derived from *Pseudomonas aeruginosa* (ntPE) can be used to transport a protein cargo (GFP) across intact intestinal epithelia. Further, we describe methods to efficiently and selectively couple ntPE onto biodegradable NPs prepared from alginate and chitosan (AC). Our results show that chemical conjugation of ntPE can dramatically enhance the capacity of AC NPs to transport across rat jejunum epithelium *in vivo*. BSA, which is of similar molecular size and has a comparable pI to ntPE, coupled similarly to AC NPs failed to elicit this effect. We also attempted to match the frequency of coupling of ntPE and BSA on NPs that were being compared for transcytosis. Thus, the concept of using ntPE to facilitate receptor-mediated vesicular transcytosis of NPs was consistent with the data obtained. It is important to point out, however, that due to uncertainties of NP stability, distribution, and access the intestinal epithelium following intraluminal injection, it is unclear how much of the administered dose reached the epithelium for uptake. Based upon these same unknowns, we also do not know if there would be linearity for lower or higher doses to reach the epithelium. Importantly, none of the tissues we examined showed any signs of overt toxicity and studies by others where alginate-chitosan NPs were administrated orally described toxicity, similar to observations reported by others [25].

While a variety of NPs have been examined for the oral delivery of biopharmaceuticals, we believe this is the first description of using a ntPE-mediated transcytosis pathway to improve the uptake efficiency of a biodegradable NP capable of carrying a macromolecular cargo across the intestinal epithelium. Our data suggests that the NPs used in these studies were sufficiently stable to compete the transcytosis process following administration into the lumen of the rat jejunum. NPs coupled to ntPE failed to show extensive retention in epithelial cells after endocytosis and did not seem to be sequestered into large lysosome-like structures within these cells [14]. This pattern suggests that the transcytosis pathway accessed by ntPE exhibited limited entry into the lysosomal degradation pathway, implying that the hijacking of a vesicular trafficking resulting in basolateral membrane targeting and not a lysosomal fate within epithelial cells. At present, it is unclear what vesicular compartments are accessed by ntPE that could direct NPs away from the typical default pathway of lysosomal destruction. In this regard, NPs made from a polyanhydride copolymer of fumaric and sebacic acids are taken up by absorptive enterocytes and have been observed within the Golgi apparatus and secretory vesicles near the lateral edge of the cells, suggesting that after enterocyte entry some types of NPs do not traffic by default to lysosomes [26]. Thus, it is possible that ntPE may provide a similar routing via the Golgi apparatus.

The scavenger receptor low-density lipoprotein-receptor-like protein 1 (LRP-1) has been shown to be involved in the endocytosis of PE, and ntPE, preceding its transcytosis across polarized epithelial cells [12]. LRP-1 is also expressed on professional antigen presenting cells located just beneath epithelial surfaces of the body, and ntPE has been previously investigated for the delivery of peptide antigens to these cells for the purposes of intranasal vaccination [21]. Some studies have shown that NPs taken up by mouse intestine can end up in CD11c^+^ cells [27]. Importantly, these previous studies showed that 20–40 nm NPs can be taken up by enterocytes, but that NPs larger than 100 nm require uptake across epithelial cells overlying Peyer’s patches [27]. Our studies have shown that NPs which are roughly twice this size (~200 nm) at the time of application into the jejunum can also undergo transcytosis across absorptive enterocytes using a mechanism accessed by ntPE and can also result in materials being taken up into CD11c^+^ cells. These 200 nm particles could represent aggregates of NPs and that during transcytosis, the size of these particles may be dynamic as they traffic through the cellular vesicular system to complete the transcytosis process.

In general, NP transcytosis in the intestine is poor due to the natural defence mechanisms of the epithelium, which maintains the body’s homeostatic state in the face of massive numbers of dietary NP-size materials and viral exposure. Several transcytosis pathways associated with receptor-mediated endocytosis mechanisms have been explored to overcome this issue: Vitamin B_12_, lectins, IgG, and bacterial toxins [6]. We have examined a specific virulence factor secreted from *Pseudomonas aeruginosa* known as exotoxin A (PE) to improve the apical to basal transport of NPs across intestinal epithelial cells. PE’s role in the pathophysiology of mucosal *Pseudomonas aeruginosa* infection has been suggested to involve local targeting of antigen presentation cells (APCs) such as macrophages and dendritic cells that would result in the ultimate apoptosis of these cells [22]. Importantly, the genetic deletion of a glutamic acid at position 553 of PE results in a completely non-toxic version of PE (ntPE) [28]. Relevant to the studies described herein, PE has been shown to efficiently transport across intact polarized epithelia to target APCs in the underlying *lamina propria* and ntPE has been shown to retain these transcytosis and APC targeting characteristics. 

Overall, we have provided evidence that the transcytosis pathway accessed by PE could be used to enhance the apical to basolateral delivery of NPs and showed that ntPE appears capable of facilitating endocytosis and transcytosis of NPs in the order of 100 nm diameter. The apparent preference for CD11c+ cells in the lamina propria following this transcytosis process suggests that ntPE-decorated NPs may have a use in oral immune system regulation: vaccination, tolerization, etc. Additionally, the AC NPs used in these studies were shown to carry a macromolecule (GFP) that could be replaced with a cargo appropriate for an oral immunization outcome.

## Figures and Tables

**Figure 1 pharmaceutics-13-01171-f001:**
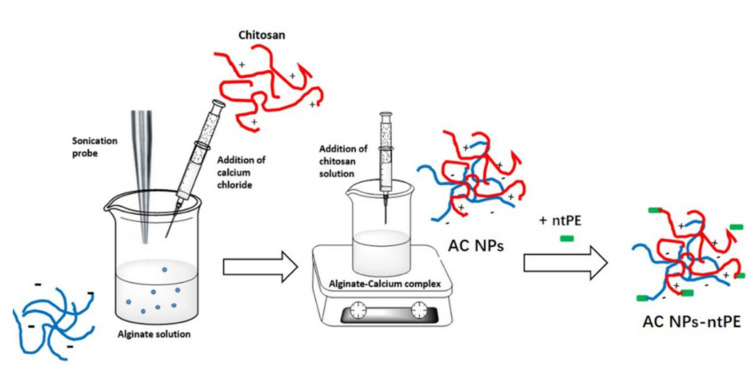
Overview illustration of alginate-chitosan (AC) nanoparticles (NPs) and stage of chemical coupling of non-toxic *Pseudomonas aeruginosa* exotoxin A (ntPE).

**Figure 2 pharmaceutics-13-01171-f002:**
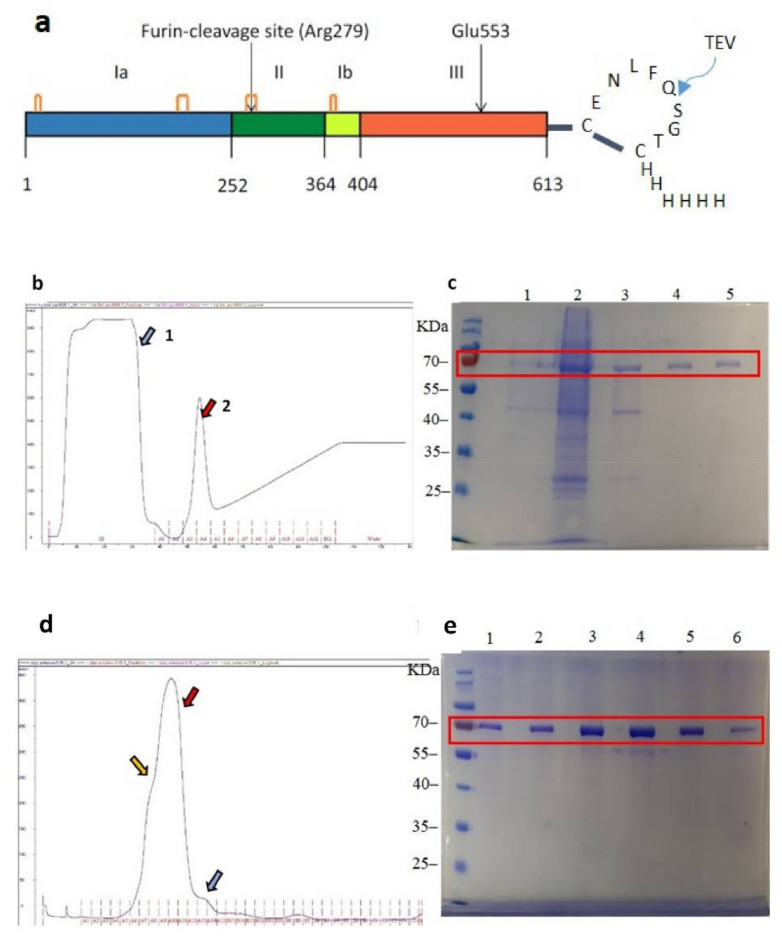
Isolation and purity assessment of ntPE-TEV-H_6_ after *E. coli* expression. (**a**) Linear diagram of ntPE-TEV-H_6_. (**b**) Enrichment of ntPE-TEV-H_6_ using HisTRAP chromatography showing column flow-through (blue arrow, (1) and imidazole-induced elution (red arrow, (2) monitored at 280 nm. (**c**) SDS-PAGE analysis: lane 1, protein from *E. coli* before the IPTG induction; lane 2, proteins from *E. coli* after IPTC induction; lane 3, protein extract. (**d**) Size exclusion chromatography of ntPE-TEV-H_6_ following HisTRAP chromatography demonstrating probable aggregated (yellow arrow, (3), monomeric (red arrow, (4), and degraded (blue arrow, (5) forms of the protein. (**e**) SDS-PAGE analysis: lanes 1–6 corresponding to collected protein extracts A7–A12, respectively. Material shown in lanes 2–5 was combined and used for conjugation to nanoparticles.

**Figure 3 pharmaceutics-13-01171-f003:**
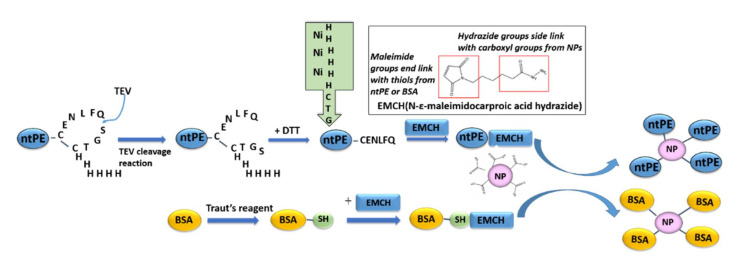
Strategy for conjugation of ntPE or BSA onto nanoparticles (NPs). Diagram of strategy for coupling through the free sulfhydryl moiety of ntPE-CENLFQ onto the surface of NPs using EMCH, followed by specific cleavage by TEV protease. BSA, as a control protein to replace ntPE, was coupled after chemical modification of amine moieties using Traut’s reagent.

**Figure 4 pharmaceutics-13-01171-f004:**
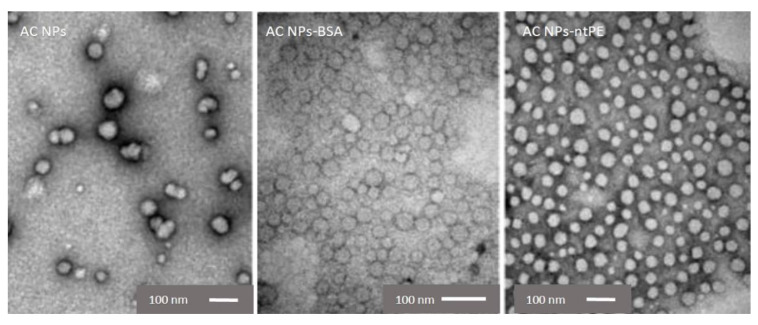
Structural examination of NPs. Transmission electron microscopy images of alginate/chitosan condensate nanoparticles (AC NPs) before and after chemical coupling with bovine serum albumin (AC NPs-BSA) or non-toxic *Pseudomonas aeruginosa* exotoxin A (AC NPs-ntPE), demonstrated by negative staining. Scale bar = 100 nm.

**Figure 5 pharmaceutics-13-01171-f005:**
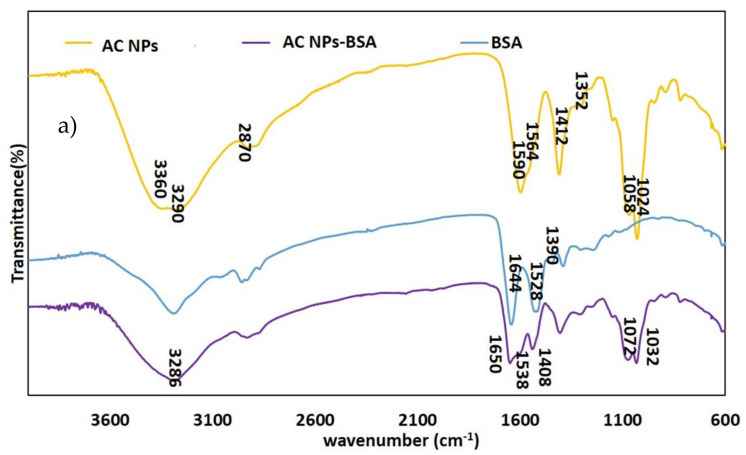
FTIR spectra of NPs. Alginate/chitosan nanoparticles (AC NPs) and bovine serum albumin (BSA) or non-toxic *Pseudomonas aeruginosa* exotoxin A (ntPE) before and after coupling to AC NPs. (**a**) Spectra obtained for AC NPs, BSA, and AC NPs-BSA. (**b**) Spectra obtained for AC NPs, ntPE, and AC NPs-ntPE.

**Figure 6 pharmaceutics-13-01171-f006:**
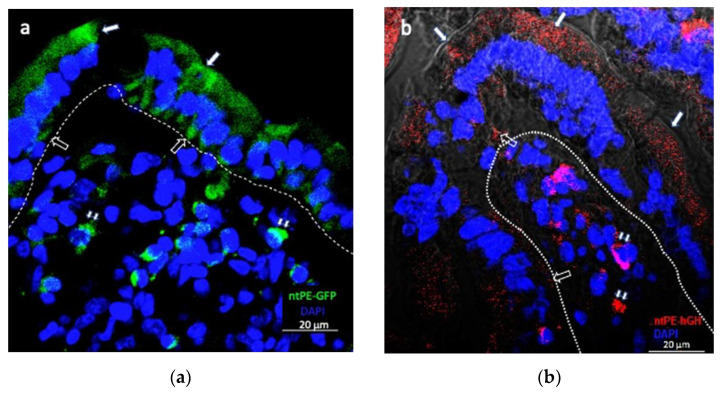
Representative confocal fluorescence image of (**a**) ntPE-GFP or (**b**) ntPE-hGH fate following intra-luminal injection (ILI) into rat jejunum. A non-toxic version of *Pseudomonas aeruginosa* exotoxin A (ntPE) genetically fused to green fluorescent protein (GFP) or human growth hormone (hGH) was examined for its tissue distribution at 15 min post-ILI. ntPE-GFP can be observed in vesicular structures within the apical region (solid arrow) and basal region of enterocytes of the epithelium. ntPE-GFP that has completed the transcytosis process can be seen in non-polarized cells within the lamina propria (small double solid arrows). Lamina propria-epithelium demarcation = dashed line. Nuclei were stained blue with DAPI. Scale bar = 10 µm.

**Figure 7 pharmaceutics-13-01171-f007:**
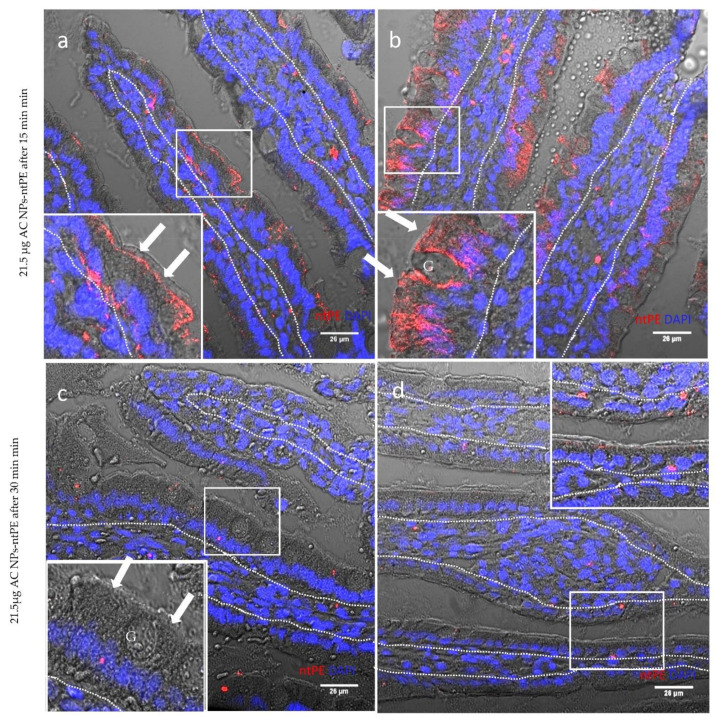
Representative confocal fluorescence images of intestinal villi following intra-luminal injection (ILI) into rat jejunum of test articles. Alginate/chitosan nanoparticles AC NPs were coupled to a non-toxic version of Pseudomonas aeruginosa exotoxin A (ntPE) that had been labeled with red fluorescent dye Alexa Fluor^®®^ 546 (AC NPs-ntPE*). Tissue distribition of AC NPs-ntPE* after 15 min post-ILI (**a**,**b**) and after 30 min post-ILI (**c**,**d**). Apical surface of enterocytes = arrow. Mucin-filled goblet cell = G. Lamina propria-epithelium demarcation = dashed line. Nuclei were stained blue with DAPI. Scale bar = 26 µm.

**Figure 8 pharmaceutics-13-01171-f008:**
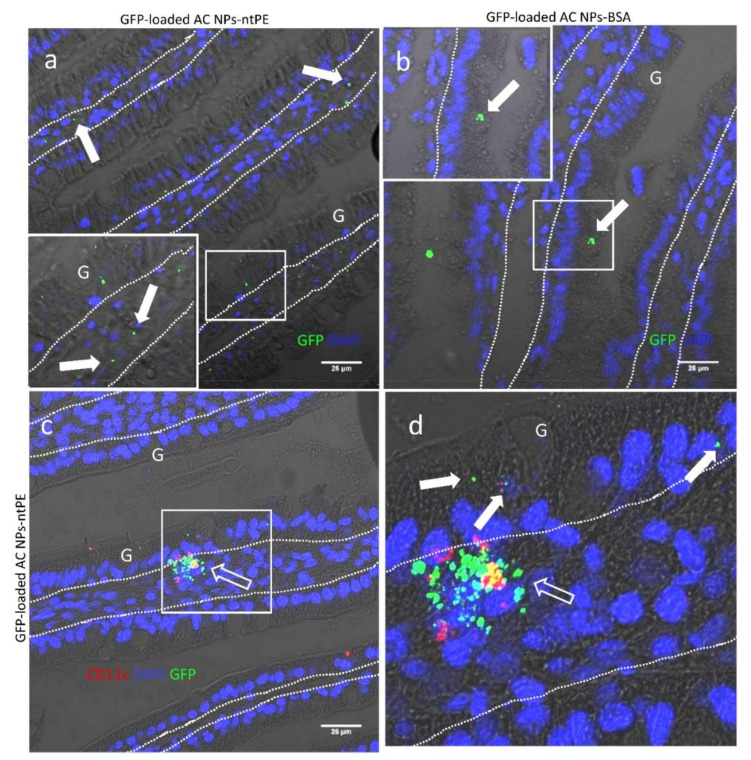
Representative confocal fluorescence images of intestinal villi following intra-luminal injection (ILI) into rat jejunum of test articles. (**a**) Green fluorescent protein (GFP)-loaded alginate/chitosan (AC NPs) coupled to a non-toxic version of Pseudomonas aeruginosa exotoxin A (AC NPs-ntPE-GFP) or (**b**) coupled to bovine serum albumin (AC NPs-BSA-GFP) at 30 min post-ILI. (**c**,**d**) AC NPs-ntPE in the villi co-localising with CD 11c^+^ cells ((**d**) is a magnification of a specific localtion identified (**c**)). GFP localizations = solid arrows. CD11c localizations = open arrows. Mucin-filled goblet cell = G. Lamina propria-epithelium demarcation = dashed line. Nuclei were stained blue with DAPI. Scale bar = 26 µm.

**Table 1 pharmaceutics-13-01171-t001:** Physicochemical characterization of nanoparticles (NPs).

Parameter	AC NPs	AC NPs-BSA	AC NPs-ntPE
Size determined from TEM ^a^	48 ± 14 nm	48 ± 15 nm	46 ± 24 nm
Size determined by NanoSight ^b^	215 ± 8 nm	280 ± 9 nm	202 ± 20 nm
Zeta potential ^b^	+20 ± 8 mV	−19 ± 11 mV	−6 ± 5 mV
Protein molecules/nanoparticle ^c^	---	3.30 ± 0.90 × 10^5^	3.41 ± 0.82 × 10^5^

^a^ Values were obtained from average of 100 individual nanoparticles. ^b^ Measurements represent mean ± SD for *n* = 3. BSA = bovine serum albumin; ntPE = non-toxic *Pseudomonas aeruginosa* exotoxin A. ^c^ Calculated as described using Formula (1) in Section 2.

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
