# Peer review of "Intestinal Transcytosis of a Protein Cargo and Nanoparticles Mediated by a Non-Toxic Form of Pseudomonas aeruginosa Exotoxin A"

_pharmaceutics, 2021, doi:10.3390/pharmaceutics13081171_

Round 1
Reviewer 1 Report
The present manuscript aims to demonstrate that biodegradable NPs decorated with non-toxic exotoxin A derived from Pseudomonas aeruginosa, present enhanced endocytosis and transcytosis capability in the intestinal epithelium (with outnumbers of enterocytes) so it can be used for oral drug/protein delivery. Therefore, the authors use an alginate/chitosan NP (described elsewhere) decorated with a non-toxic form of the exotoxin A (here named nt-PE) and injected in the lumen of male rat jejunums. Authors used the same biodegradable NPs but decorated with BSA as negative control for the transcytosis capacity. Then, jejunum samples were taken after 15 and 30 min and imaged using confocal microscopy to determine the transcytotic capability of both NPs.
This is an original manuscript studying new strategies for drug delivery using NPs in combination with enhancers to solve the poor or unspecific internalization. The manuscript is concise and understandable. However, further efforts should be considered to improve the quality of the images presented as well as the addition of results regarding BSA results.
Major comments
MATERIAL AND METHODS
P3. Were NPs measured randomly from different TEM images and how were images (indicate in section 2.6 of material and methods)? How many biological and/or technical replicates?
P4, section 2.8. Why were NPs injected at a concentration of 86 ug/mL? Was there a previous toxicological assay performed? Is this number derived from previous studies? This should be indicated.
RESULTS
P6. Was the lifetime of the biodegradable AC NPs evaluated in vivo or in vitro?
P6. Since the SD of primary particles measurements (48±14, 48±15 and 46±24) is considerable high, I suggest plotting a NPs frequency diagram to see the size distribution. Why do the authors talk about “slight agglomeration” of AC NPs? Is this statement based only in TEM images? Is this a trend in other images (if there are)?
P6 and 7. Authors should talk about “primary particles” when referring to those dry NPs from TEM images and about hydrodynamic size/diameters of NPs when talking about hydrated samples measured by DLS.
“The Nanoparticle Tracking Analysis (NTA) software analyses many particles individually and simultaneously (particle-by-particle), and by using the Stokes Einstein equation, calculates their hydrodynamic diameters.” (Malvern Panalytical)
P7. Was the increment or reduction of the hydrodynamic size of NPs (compared to AC NPs) statistically significant?
P7. AC NPs-ntPE present a very little stability in solution (-6 ±5mV), which usually means aggregation as well as agglomeration when hydrated. Do the authors have any strategy for a better NPs dispersion? Is the particle stability known to affect its cell internalization, biodistribution along the surface or kinetics?
P8. Peaks between 1020 and 1030 of Figure 4 could be further mentioned, described and/or discussed. See the following reference as an example: Kyzioł A, Mazgała A, Michna J, Regiel-Futyra A, Sebastian V. Preparation and characterization of alginate/chitosan formulations for ciprofloxacin-controlled delivery. J Biomater Appl. 2017 Aug;32(2):162-174. doi: 10.1177/0885328217714352. Epub 2017 Jun 26. PMID: 28649925.
P8/9 Figure 5. How do the authors differentiate between enterocytes and goblet cells? The arrow, the G, and dashed line from the Figure 5 caption does not appear in any picture. Moreover, there are red artifacts behind the scale bar of every picture.
P9. L296. How are the authors sure that the AC NPs-ntPE is crossing the intestinal epithelium intracellularly but not paracellularly as well? The bright field of this images do not clearly elucidate the cells membrane perimeters from the cell-cell interspace to ensure the exact location of those Alexa fluor labeled NPs. Further staining or dyes are suggested to (1) differentiate between cell types (e.g. Ab anti-villi proteins, anti-filaments of actin, SGLT1, GLUT-2, anti-mucosubstances like WGA, etc), and (2) localize cell membrane and delineate cell-cell interspace (e.g. ZO-1, OCCL, CLDN, cellMask, etc).
P9. How do the authors explain that after 30 min the red signal coming from AC NPs-ntPE have diminished that much? An intermediate time point between 15 and 30 min after ILI administration would be necessary to visualize the NPs deeper penetration to the lamina propria before vanishing. Moreover, images at time point 0 would be useful to demonstrate that the ILI was successfully conducted, and all NPs were placed ONLY in the lumen compartment.
Have the authors characterized/imaged the AC NPs-ntPE* and ntPE* alone by confocal microscopy?
P9. Images or results regarding the work with AC NPs-BSA* (as supplementary information) would further clarify and support the statement that authors make in lines 304 to 308 and lines 353 to 354 about non-Specific uptake.
Why is not the fluorescence of GFP protein shown in Figure 5 to see the co-localization between AC NPs-ntPE* and its cargo? And why there is no images of samples after 15 min injection in Figure 6 if previously has been demonstrated the abundance of A C NPs-ntPE* penetrating the barrier?
DISCUSSION
A larger discussion about the life spam of the biodegradable AC NPS-ntPE in the intestinal epithelium would be suitable to understand the pros and cons of its utilization as oral drug delivery.
P12, L382-385. Presenting such a low zeta potential value for your AC NPs-ntPE samples, is very risky to confirm that those NPs have a size around 200 nm instead of talking about aggregated NPs.
Is there any investigation regarding AC NPs-ntPE or the exotoxin and in vitro models (like Caco-2 monolayers, Caco-2/HT29-MTX barriers, etc) of the intestinal tract to correlate this findings and further discuss?
Minor comments
L 11, 67, 240 and 338. Pseudomonas aeruginosa
P4, L 153. Test Particles
P4, L158. 250 µL
Figure 1. What are the lanes 4 and 5 from SDS image “C” in figure 1?
Figure 1. Images D and E are overlapping.
Figure 4. Maintain the legend on both graphs (a and b) consistent with the manuscript. AC NPs, AC NPs-BSA, AC NPs-ntPE instead of Alg-Chit-NPs or NPs-PE …
P10, L308. Lacking a dot at the end of the sentence.
P10, L326. “Within the” is repeated.
Figure 5 and 6 needs to be edited as seems that everything has moved in the process.
Figure 5 and 6. I am not sure about test articles or tested particles?
L 19, 113, 296, 311 and 315. In vivo
Author Response
Reviewer 1
Major comments
MATERIAL AND METHODS
P3. Were NPs measured randomly from different TEM images and how were images (indicate in section 2.6 of material and methods)? How many biological and/or technical replicates?
Our response/action(s) –
At least three batches of each NP were prepared and observed under TEM. Representative photos presented in the manuscript were taken from batches prepared on the same date to optimize comparisons/characterizations of NPs. Average NP size was determined from the measurement of 100 randomly selected NPs from different locations on the samples. We have added information in the Methods text.
P4, section 2.8. Why were NPs injected at a concentration of 86 ug/mL? Was there a previous toxicological assay performed? Is this number derived from previous studies? This should be indicated.
Our response/action(s) – The goal of these studies focused on assessing the transport capability enabled by nt-PE. 86 ug/mL merely turned out to be a convenient amount of material at the end of the production process that could be readily tested in vivo. No study to examine other doses was performed. Due to uncertainties of NP stability, distribution, and access the intestinal epithelium following intraluminal injection, it is unclear how much of this dose reached the epithelium for endocytosis and transcytosis. Based upon these same unknowns, we also do not know if there would be linearity for other injected doses to reach the epithelium. None of the tissues we examined showed any signs of overt toxicity. Studies by others, where alginate-chitosan NPs were administrated orally, have not described any toxicity. We have added a reference and text in the Discussion section.
RESULTS
P6. Was the lifetime of the biodegradable AC NPs evaluated in vivo or in vitro?
Our response/action(s) – We did perform some in vitro studies regarding in vitro stability and this information is now included. Stability of this ntPE-polymer conjugate was sufficiently stable (over hours) top acquire size, zeta potential, and FTIR information. Thus, these particles were considered to be sufficiently stable for the 15-30 minutes required for uptake across the intestinal epithelium following intraluminal injection in rats. NP biodegradation after epithelial transcytosis would be dependent upon a variety of degradative processes in cells within the intestinal lamina propria following transcytosis. We have added text in the Results and Discussion sections to point out these issues.
P6. Since the SD of primary particles measurements (48±14, 48±15 and 46±24) is considerable high, I suggest plotting a NPs frequency diagram to see the size distribution. Why do the authors talk about “slight agglomeration” of AC NPs? Is this statement based only in TEM images? Is this a trend in other images (if there are)?
P6 and 7. Authors should talk about “primary particles” when referring to those dry NPs from TEM images and about hydrodynamic size/diameters of NPs when talking about hydrated samples measured by DLS. “The Nanoparticle Tracking Analysis (NTA) software analyses many particles individually and simultaneously (particle-by-particle), and by using the Stokes Einstein equation, calculates their hydrodynamic diameters.” (Malvern Panalytical)
Our response/action(s) – We agree and have added information in the Results text.
P7. Was the increment or reduction of the hydrodynamic size of NPs (compared to AC NPs) statistically significant?
Our response/action(s) – We calculated the statistical significance between the difference AC NPs and added information in the Results text.
P7. AC NPs-ntPE present a very little stability in solution (-6 ±5mV), which usually means aggregation as well as agglomeration when hydrated. Do the authors have any strategy for a better NPs dispersion? Is the particle stability known to affect its cell internalization, biodistribution along the surface or kinetics?
Our response/action(s) – We agree that zeta potential (electrostatic interaction) can be related to stability, but also point out that stability of colloids is controlled by other forces, such as van der Waals attractions. We did not observe dramatic instability in this NP preparation as predicted by the reviewer. Similar to our data, others have noted that proteins decorated on the surface of NPs can increase their colloidal stability (Moore T, Rodriguez-Lorenzo L, et al. Nanoparticle colloidal stability in cell culture media and impact on cellular interactions). We mention the idea of colloidal stability in the Results section.
As we did not observe overt issues of stability and aggregation for AC NPs-ntPE or AC NPs-BSA in the preparations, there was no need to describe a strategy to increase NP dispersion. This is now addressed in the Results text.
P8. Peaks between 1020 and 1030 of Figure 4 could be further mentioned, described and/or discussed. See the following reference as an example: Kyzioł A, Mazgała A, Michna J, Regiel-Futyra A, Sebastian V. Preparation and characterization of alginate/chitosan formulations for ciprofloxacin-controlled delivery. J Biomater Appl. 2017 Aug;32(2):162-174. doi: 10.1177/0885328217714352. Epub 2017 Jun 26. PMID: 28649925.
Our response/action(s) – Thank you for recommending this reference. The main purpose for showing FTIR spectra was to confirm the linkage of protein (ntPE and BSA) on the NPs. This reference helped us to confirm the presence of alginate as the peaks between 1020 and 1030 indicated. We have added text in the Results section and included this reference.
P8/9 Figure 5. How do the authors differentiate between enterocytes and goblet cells? The arrow, the G, and dashed line from the Figure 5 caption does not appear in any picture. Moreover, there are red artifacts behind the scale bar of every picture.
Our response/action(s) – Thank you for bringing this to our attention. There was a problem with the transfer of image text. This is now corrected. Goblet cells, that contain mucin which is lost during tissue processing, have a very distinctive morphology compare to enterocytes. The red artifacts behind the scale bar supposed to be “ntPE”, which indicating that the red signals are from NPs.
It meant to look like this and has now been corrected.
P9. L296. How are the authors sure that the AC NPs-ntPE is crossing the intestinal epithelium intracellularly but not paracellularly as well? The bright field of this images do not clearly elucidate the cells membrane perimeters from the cell-cell interspace to ensure the exact location of those Alexa fluor labeled NPs. Further staining or dyes are suggested to (1) differentiate between cell types (e.g. Ab anti-villi proteins, anti-filaments of actin, SGLT1, GLUT-2, anti-mucosubstances like WGA, etc), and (2) localize cell membrane and delineate cell-cell interspace (e.g. ZO-1, OCCL, CLDN, cellMask, etc).
Our response/action(s) – We cannot rule out that paracellular leak could account for some NPs crossing the intestinal epithelia. Indeed, we are careful not to state that paracellular transport is “not” occurring. None of our many microscopic observations, however, support such an event. The cell membranes are indeed sufficiently distinct to clearly shown that transporting NPs are in cytoplasmic compartment of enterocytes and goblet cells. These images are consistent with published images of this class of toxins transporting across intestinal epithelium via a transcytosis process (Tissue Barriers. 2020;8(1):1710429. doi: 10.1080/21688370.2019.1710429. Epub 2020 Jan 13.PMID: 31928299) that was referenced in the manuscript.
P9. How do the authors explain that after 30 min the red signal coming from AC NPs-ntPE have diminished that much? An intermediate time point between 15 and 30 min after ILI administration would be necessary to visualize the NPs deeper penetration to the lamina propria before vanishing. Moreover, images at time point 0 would be useful to demonstrate that the ILI was successfully conducted, and all NPs were placed ONLY in the lumen compartment.
Our response/action(s) – Exotoxin transcytosis across polarized epithelial cells occurs in 10-15 min (Tissue Barriers. 2020;8(1):1710429. doi: 10.1080/21688370.2019.1710429. Epub 2020 Jan 13.PMID: 31928299). During this transcytosis, the exotoxin, and presumably whatever it is bring along as a cargo, are kept out of highly degrading environments such as lysosomes. Following this transcytosis process and uptake into non-polarized cells in the lamina propria, however, the exotoxoin and presumably whatever it is bring along as a cargo are trafficking to lysosomes (Tissue Barriers. 2020;8(1):1710429. doi: 10.1080/21688370.2019.1710429. Epub 2020 Jan 13.PMID: 31928299). Thus, AC NPs-ntPE should have a very short half-life in the lamina propria and the difference in detection between 15 min when transcytosis is still occurring and 30 min when most of the transported material has enter into cells within the lamina propria is not surprising. We have added Results text to provide an explanation for this rapid decline in detectable fluorescence.
A true T=0 time point is unrealistic in this ILI administration model. Our animal license requires a euthanasia protocol involving a CO2 chamber for 5-10 mins. This is why we picked 15 min as our first data time point.
Have the authors characterized/imaged the AC NPs-ntPE* and ntPE* alone by confocal microscopy?
Our response/action(s) – Others have shown biotin-labeled ntPE to transport across epithelia in a manner similar to what we have observed for the AC NPs-ntPE* (Ref 19). We have added Results text related to this. This does not address the concern, however, of macromolecular cargo being transported by ntPE across intestinal epithelial cells. We have included a new figure that shows the efficient transcytosis of ntPE genetically coupled to human growth hormone following ILI administration. This data highlights the transcytosis, rather than paracellular, pathway taken by this exotoxin to access cells within the lamina propria following application to the apical intestinal surface.
P9. Images or results regarding the work with AC NPs-BSA* (as supplementary information) would further clarify and support the statement that authors make in lines 304 to 308 and lines 353 to 354 about non-Specific uptake.
Our response/action(s) We have included data for the GFP-loaded AC NPs-BSA materials (Fig. 6b) as a demonstration for the BSA control. We feel that this encapsules the best control for the question that is being asked in these studies: can AC NPs decorated with ntPE deliver a macromolecule to cells in the lamina propria following application at the intestinal epithelial surface? Thus, we feel that the statements made in these lines are warranted by the data presented.
Why is not the fluorescence of GFP protein shown in Figure 5 to see the co-localization between AC NPs-ntPE* and its cargo? And why there is no images of samples after 15 min injection in Figure 6 if previously has been demonstrated the abundance of A C NPs-ntPE* penetrating the barrier?
Our response/action(s) – Figure 5 examined the transport of AC NPs-ntPE* that did not contain GFP. Figure 6 focused on GFP cargo delivery and used AC NPs that lacked a fluorescent tag on ntPE. We were only interested in the potential accumulation of the NP cargo (GPF) into cells within the lamina propria, so we focused on this later time point. We have added Results text related to this point.
DISCUSSION
A larger discussion about the life spam of the biodegradable AC NPS-ntPE in the intestinal epithelium would be suitable to understand the pros and cons of its utilization as oral drug delivery.
Our response/action(s) – There is really no way to determine the life span of these particles in the lamina propria as we are not sure of any method to faithfully collect and analyze them without compromising their stability in the process. The use of such a system can only be truly assessed by an associated PD outcome, and that will be dependent upon the nature of the cargo to be delivered. Thus, we do not feel that we can say anything meaningful in such a discussion.
P12, L382-385. Presenting such a low zeta potential value for your AC NPs-ntPE samples, is very risky to confirm that those NPs have a size around 200 nm instead of talking about aggregated NPs.
Our response/action(s) – The fact that we observe AC NPs-ntPE* in epithelial cells in the size range of 200 nm and not large aggregates suggests that if such an aggregation event occurred, it was not extensive and did not preclude the correct size particles from entering into epithelial cells.
Is there any investigation regarding AC NPs-ntPE or the exotoxin and in vitro models (like Caco-2 monolayers, Caco-2/HT29-MTX barriers, etc) of the intestinal tract to correlate this findings and further discuss?
Our response/action(s) – Epithelial transport across these in vitro cell systems have been described and we have incorporated these studies in the references. Drug Discov Today. 2002 Feb 15;7(4):247-58. doi: 10.1016/s1359-6446(01)02139-0.PMID: 11839522
Minor comments – all fixed in text
L 11, 67, 240 and 338. Pseudomonas aeruginosa
P4, L 153. Test Particles
P4, L158. 250 µL
Figure 1. What are the lanes 4 and 5 from SDS image “C” in figure 1?
Figure 1. Images D and E are overlapping.
Figure 4. Maintain the legend on both graphs (a and b) consistent with the manuscript. AC NPs, AC NPs-BSA, AC NPs-ntPE instead of Alg-Chit-NPs or NPs-PE …
P10, L308. Lacking a dot at the end of the sentence.
P10, L326. “Within the” is repeated.
Figure 5 and 6 needs to be edited as seems that everything has moved in the process.
Figure 5 and 6. I am not sure about test articles or tested particles?
L 19, 113, 296, 311 and 315. In vivo
Reviewer 2 Report
Overall this is a high quality paper and a well described and executed study. My critiques are minor in nature.
I recommend the addition of a schematic of the chemistry/synthesis of nanoparticles. Change the '-1' to a superscript in FTIR section, italicize 'in vivo' throughout, there's an odd character before "L" in the 6th line of section 2.8. Formatting of figure captions was confusing until I realized the real caption was the paragraph under the figure title, which seems an odd formatting choice. Table 1, superscript c reads 'calculated as described', which gives the reader no useful information and should either be reworded to include experimental detail or the equation number or simply removed.
Is there a way to quantify the data from the in vivo studies that indicates the percentage of particles that are taken up or transcytosed? THis would be powerful in convincing the reader that this is an effective particle coating protein for trancytosis.
A discussion of the expected biodistribution of the particles would be helpful, since those studies weren't done here as only short time periods were studied. What role do you anticipate the protein corona playing in the ultimate fate of these particles?
Author Response
Reviewer 2
I recommend the addition of a schematic of the chemistry/synthesis of nanoparticles.
Our response/action(s) – We have produced a graphical abstract with this information.
Change the '-1' to a superscript in FTIR section
Our response/action(s) – Done
italicize 'in vivo' throughout
Our response/action(s) – Done
there's an odd character before "L" in the 6th line of section 2.8.
Our response/action(s) – Done
Formatting of figure captions was confusing until I realized the real caption was the paragraph under the figure title, which seems an odd formatting choice
Our response/action(s) – We have made changes to the figure legends to reduce confusion.
Table 1, superscript c reads 'calculated as described', which gives the reader no useful information and should either be reworded to include experimental detail or the equation number or simply removed.
Our response/action(s) – Done
We used this equation to calculate, it’s mentioned in the method section
Is there a way to quantify the data from the in vivo studies that indicates the percentage of particles that are taken up or transcytosed? THis would be powerful in convincing the reader that this is an effective particle coating protein for trancytosis. –
Our response/action(s) – As above for the other reviewer. There is really no way to determine the extent of these particles reaching the lamina propria as we are not sure of any method to faithfully collect and analyze them without compromising their stability in the process. The use of such a system can only be truly assessed by an associated PD outcome, and that will be dependent upon the nature of the cargo to be delivered. Thus, we do not feel that we can say anything meaningful in such a discussion.
A discussion of the expected biodistribution of the particles would be helpful, since those studies weren't done here as only short time periods were studied. What role do you anticipate the protein corona playing in the ultimate fate of these particles?
Our response/action(s) – The focus of these studies was to merely show that transport of a macromolecule cargo in biodegradable particle can be facilitated by ntPE. Any meaningful outcome will be cargo dependent, with the most useful outcome being related to PD rather than a PK-type measure since all elements in the system are intended to be labile.
Round 2
Reviewer 1 Report
Although most of the revisions were answered and suggestions considered, there are some remaining:
- The following question was not answered:
P6. Since the SD of primary particles measurements (48±14, 48±15 and 46±24) is considerably high, I suggest plotting a NPs frequency diagram to see the size distribution. Why do the authors talk about the “slight agglomeration” of AC NPs? Is this statement based only on TEM images? Is this a trend in other images (if there are)?
2. Please make sure to italicize all Pseudomonas aeruginosa present in the manuscript.
3. Double check sentences from lines 425-431, there is repeated content.
4. P12, L382-385. Presenting such a low zeta potential value for your AC NPs-ntPE samples is very risky to confirm that those NPs have a size around 200 nm instead of talking about aggregated NPs.
Our response/action(s) – The fact that we observe AC NPs-ntPE* in epithelial cells in the size range of 200 nm and not large aggregates suggests that if such an aggregation event occurred, it was not extensive and did not preclude the correct size particles from entering into epithelial cells.
Regarding the statement above: 1. the NPs size of 200 nm was measured in the suspension solution and not in the epithelial cells (which also could change due to biological interactions), then a more accurate statement would be to say: the fact that we inject NPs in the size range 200 nm...: and 2. the previous suggestion was not clearly answered. Based on the data from table 1, how would the authors explain (if not by an aggregation event) that in its dry form, NPs measure 46 +/- 24 nm, but hydrated the size increase up to 202 nm? Again, the fact to have a low zeta potential, which means poor sample stability, explains that aggregation events are highly probable to happen. My suggestion it that would be more accurate to talk about aggregates of NPs entering in epithelial cells rather than NPs of 200 nm.
Author Response
210715 Second review by reviewer 1
- The following question was not answered:
P6. Since the SD of primary particles measurements (48±14, 48±15 and 46±24) is considerably high, I suggest plotting a NPs frequency diagram to see the size distribution. Why do the authors talk about the “slight agglomeration” of AC NPs? Is this statement based only on TEM images? Is this a trend in other images (if there are)?
Our response: To simplify this issue, we have removed the statement about agglomeration and have altered the text accordingly.
- Please make sure to italicize all Pseudomonas aeruginosa present in the manuscript.
Our response: Done
- Double check sentences from lines 425-431, there is repeated content.
Our response: The two sentences focus on two aspects of the same issue. The first is that it is very difficult to obtain accurate quantitative outcomes due to uncertainties of how much intact material actually reached the epithelial surface due to the factors listed. This, in turn, limits the potential to assume linearity of outcomes for these applications. Thus, we do not believe there is repeated content.
- P12, L382-385. Presenting such a low zeta potential value for your AC NPs-ntPE samples is very risky to confirm that those NPs have a size around 200 nm instead of talking about aggregated NPs.
Our response/action(s) – The fact that we observe AC NPs-ntPE* in epithelial cells in the size range of 200 nm and not large aggregates suggests that if such an aggregation event occurred, it was not extensive and did not preclude the correct size particles from entering into epithelial cells.
Regarding the statement above: 1. the NPs size of 200 nm was measured in the suspension solution and not in the epithelial cells (which also could change due to biological interactions), then a more accurate statement would be to say: the fact that we inject NPs in the size range 200 nm...: and 2. the previous suggestion was not clearly answered. Based on the data from table 1, how would the authors explain (if not by an aggregation event) that in its dry form, NPs measure 46 +/- 24 nm, but hydrated the size increase up to 202 nm? Again, the fact to have a low zeta potential, which means poor sample stability, explains that aggregation events are highly probable to happen. My suggestion it that would be more accurate to talk about aggregates of NPs entering in epithelial cells rather than NPs of 200 nm.
Our response: We have added discussion text as suggested.
Round 3
Reviewer 1 Report
Minor revisions were done. No further comments.